# Explain Breathlessness: Could ‘Usual’ Explanations Contribute to Maladaptive Beliefs of People Living with Breathlessness?

**DOI:** 10.3390/healthcare12181813

**Published:** 2024-09-10

**Authors:** Marie T. Williams, Hayley Lewthwaite, Dina Brooks, Kylie N. Johnston

**Affiliations:** 1Innovation, IMPlementation and Clinical Translation in Health (IIMPACT), University of South Australia, Adelaide, SA 5000, Australia; hayley.lewthwaite@newcastle.edu.au (H.L.); kylie.johnston@unisa.edu.au (K.N.J.); 2Centre of Research Excellence in Asthma Treatable Traits, College of Health, Medicine and Wellbeing, University of Newcastle, Newcastle, NSW 2308, Australia; 3Asthma and Breathing Research Program, Hunter Medical Research Institute, Newcastle, NSW 2305, Australia; 4Hamilton and West Park Health Care Centre, School of Rehabilitation Sciences, McMaster University, Hamilton, ON L8S 4L8, Canada; brookd8@mcmaster.ca

**Keywords:** chronic breathlessness, dyspnea, education, beliefs, expectations, communication

## Abstract

Background: Explanations provided by healthcare professionals contribute to patient beliefs. Little is known about how healthcare professionals explain chronic breathlessness to people living with this adverse sensation. Methods: A purpose-designed survey disseminated via newsletters of Australian professional associations (physiotherapy, respiratory medicine, palliative care). Respondents provided free-text responses for their usual explanation and concepts important to include, avoid, or perceived as difficult to understand by recipients. Content analysis coded free text into mutually exclusive categories with the proportion of respondents in each category reported. Results: Respondents (*n* = 61) were predominantly clinicians (93%) who frequently (80% daily/weekly) conversed with patients about breathlessness. Frequent phrases included within usual explanations reflected breathlessness resulting from medical conditions (70% of respondents) and physiological mechanisms (44%) with foci ranging from multifactorial to single-mechanism origins. Management principles were important to include and phrases encouraging maladaptive beliefs were important to avoid. The most frequent difficult concept identified concerned inconsistent relationships between oxygenation and breathlessness. Where explanations included the term ’oxygen’, a form of cognitive shortcut (heuristic) may contribute to erroneous beliefs. Conclusions: This study presents examples of health professional explanations for chronic breathlessness as a starting point for considering whether and how explanations could contribute to adaptive or maladaptive breathlessness beliefs of recipients.

## 1. Introduction

People living with persistent disabling symptoms, such as chronic breathlessness, and health professionals involved in their care develop explanations as a way of ‘making sense’ of symptoms. Chronic, disabling breathlessness is a complex, common symptom across a range of cardiorespiratory and neuromuscular chronic conditions. While almost always pathophysiological in origin, an individual’s conscious awareness of chronic breathlessness is not a simple direct reflection of the biological impairments underpinning an individual’s chronic health condition. Multiple factors contribute to the symptom experience of breathlessness (e.g., learned associations, beliefs, expectations, coping strategies, anxiety, mood and affect, current fitness, environmental conditions) [1,2]. Many of these factors have effective evidence-based interventions capable of palliating breathing distress, which can be applied in addition to, or independently from, disease-specific therapeutic interventions.

A health professional’s understanding, beliefs, and expectations of a clinical condition can profoundly influence patient beliefs, expectations, and behaviors [3,4,5]. Symptom explanations have the potential to facilitate adaptive beliefs (‘something can be done’), positive coping strategies, and engagement in health management, or alternatively, reinforce non-helpful maladaptive beliefs (‘nothing can be done’) and detrimental coping strategies [6,7]. Explanations are, however, challenging when a symptom such as chronic breathlessness does not have a consistent, proportional relationship with markers of disease severity (e.g., pulmonary function, imaging results, oxygenation [8]).

The need for better education specific to chronic breathlessness has been identified by people living with chronic breathlessness, informal carers, and healthcare professionals [9,10,11,12,13,14,15]. Explanations are a fundamental component of breathlessness education. Common misconceptions/maladaptive beliefs about this symptom include the following: (1) the severity of lung or heart impairment is the sole contributor to this sensation; (2) the distress caused by this sensation should be consistent with biomarkers (pulmonary/cardiac function, oxygen saturation, etc.); (3) exertional breathlessness causes harm and should be avoided; (4) this sensation will inevitably progress and worsen; (5) nothing further can be done once management is optimized (e.g., pharmacological, pulmonary rehabilitation) [16,17,18]. However, people living with breathlessness and clinicians may be reluctant to initiate conversations about breathlessness during clinical consultations [19]. Even when people living with breathlessness seek medical advice for this symptom, the advice received may not be helpful [20] or clinicians may actively avoid discussing this symptom due to a perceived inability to offer effective palliation, lack of specific training, or awareness of appropriate services [21,22].

Compared to explanations for persistent pain [23], it is difficult to find primary studies that focus on how health professionals explain chronic breathlessness to people living with this symptom. This evidence base is fragmentary. Where studies focus upon the content of explanations, these tend to reflect medically unexplained symptoms which may include breathlessness [24]. Within studies of educational interventions for breathlessness, little specific detail is reported detail for explanations [25]. Within studies reporting patient experience, verbatim exemplars from recipients of breathlessness explanations expressing dissatisfaction [26,27] are more common than those evidencing satisfaction [28].

The primary aim of this study was to describe the content of a sample of ‘usual’ explanations provided by health professionals, when explaining chronic breathlessness to people living with this symptom. Secondary aims included describing health professional perceptions of information important to include or avoid within explanations and concepts perceived to be difficult to understand by recipients.

## 2. Materials and Methods

This cross-sectional study used an anonymous purpose-designed electronic survey to collect examples of ‘usual’ explanations provided to people living with chronic breathlessness, and perspectives of explanation content from Australian health professionals. Ethical approval was provided by the University of South Australia’s Human Research Ethics Committee (No. 200896).

Participants and recruitment: Health professionals from a variety of disciplines (doctors, nurses, physiotherapists, etc.) likely to be involved in the clinical management of people living with chronic breathlessness were recruited via electronic survey links in professional newsletters (Thoracic Society of Australia and New Zealand, Australian Physiotherapy Association Cardiorespiratory Group) or professional group email communications (Queensland Cardiorespiratory Physiotherapy Network [QCRPN], Palliative Care Clinical Studies Collaborative [PaCCSC]). Within recruitment materials and the survey proper, chronic breathlessness was explicitly defined as ‘*breathlessness that persists despite optimal treatment of underlying pathology and results in disability’* [29] within a context of a common daily experience for people living with a range of malignant and non-malignant respiratory, cardiovascular, and neuromuscular conditions.

Survey development and items: The survey was prospectively planned, and pilot tested by research team members locally and residing outside of Australia for access, clarity, wording, and time to complete. The final survey comprised three sections: (1) study information, including consent processes; (2) responder demographics (Questions 1–3; professional discipline, role, and frequency of breathlessness explanations in the past 6 months); and (3) a series of open-ended questions focused upon breathlessness explanations. No clinical scenario or clinical condition/diagnosis were provided to frame questions concerning breathlessness explanations.

Respondents were invited to provide a ‘usual’ explanation (Question 4; “*If you were asked “Why am I breathless?” by a person with a long-standing medical condition where breathlessness was a daily feature, what would you normally say? We are interested in how you normally explain chronic breathlessness to a person with this symptom during a routine encounter when the person is in a stable state regardless of whether this is the first time the person has asked the question or not. In this text box, please write the explanation you would normally provide. You will have the opportunity to clarify and alter your response for a person with a specific condition in the next question*”). This item was followed by a clarifying question “*Would you change anything in your explanation if the person had a specific chronic condition such as chronic obstructive pulmonary disease, heart failure, idiopathic pulmonary fibrosis, cystic fibrosis, neuromuscular disease etc.?* (Question 5 Yes/No option). Where a respondent indicated ‘Yes’, they were provided with an option to provide clarifying information (“*If yes, what changes would you make to your explanation depending upon the person’s chronic condition?” Question 6).* Three further open-ended text items completed Section 3 (Question 7 “*What do you think are the most important points to convey when explaining chronic breathlessness to a person with this symptom?”, Question 8 “Is there any information that you avoid or try to not to say when explaining chronic breathlessness to a person with this symptom?*” (if yes, provide the points) and Question 9 “*When explaining chronic breathlessness to people living with this symptom, are there particular ideas or concepts that you think people find difficult to understand or follow?*”).

Data management: Participant demographic data were analyzed descriptively. Open-text responses were synthesized using the coding framework developed for a previous Delphi consensus process of international experts for concepts important to include or avoid when explaining chronic breathlessness to a person living with this adverse symptom (details of the coding process, including final domain and items/concepts included within each domain are available in Williams et al. (2020) [18]). The framework included codes for unique mutually exclusive concepts grouped under a domain label. In summary, each text response for open-ended questions (usual explanation, important to include/avoid, and difficult concepts) was reviewed and specific words/phrases within each text response were coded (content analysis) using the coding framework (each phrase assigned a single code only). A text response could therefore reflect several domains. The text was analyzed for content with the intent to keep as close to the descriptive phrase without interpretation, inference, or further abstraction. Verbatim responses to each of the three open-ended questions were reviewed and provisionally coded by a single researcher (MTW). Independent researchers (HL, DB, KJ) reviewed the coded statements and allocated each to a domain, with discrepancies resolved by discussion. For each open-ended question, specific codes and domains were summarized and reported for the frequency of occurrence (n=) and percentage (%) of respondents.

## 3. Results

Surveys were disseminated between February and July 2018. Of the 91 surveys submitted, 30 contained no data beyond consent (*n* = 3) or demographic characteristics (*n* = 27), leaving 61 surveys for analysis. Respondent characteristics are presented in Table 1. Respondents reflected a range of professional disciplines (physiotherapy 54%, nursing 28%, medicine 16%, respiratory science 2%). Most respondents had direct clinical contact with patients over the previous six months (93% overall, 77% predominately clinical roles, and 16% mix of clinical/academic or administrative roles) and had been involved in a conversation with a patient about chronic breathlessness at least once a day (34%), week (46%), or month (16%).

### 3.1. If You Were Asked “Why Am I Breathless?” By a Person with a Long-Standing Medical Condition Where Breathlessness Was a Daily Feature, What Would You Normally Say?

Examples of ‘usual’ explanations of chronic breathlessness were provided by fifty-seven (93%) respondents, with four respondents unable to provide an explanation without a patient diagnosis. Figure 1 presents a summary of ‘usual explanation’ content domains (Details provided in Appendix A). The most frequent statements reflected breathlessness arising from a medical condition with 40 (70%) respondents including one or both phrases (or similar) “*Due to your underlying pathology/medical condition*” or “*Because your lungs/heart are not working as well as they should”.* Specific examples of disease-specific physiological mechanisms were included by 25 respondents (44%) with the most frequent phrases reflecting breathlessness as a corollary of increased effort/work of breathing required to ensure sufficient oxygen/fuel for the body (16% of explanations). Less than a third of respondents’ explanations included the terms ‘multifactorial’, ‘multiple factors’, or ‘caused by lots of factors’” (30% of respondents) or included examples of non-disease-specific factors contributing to breathlessness (25% of respondents). Few explanations included phrases indicating inconsistent relationships between breathlessness and disease severity or oxygen levels (5% of respondents, Table 2).

### 3.2. Would You Change Anything in Your Explanation If the Person Had a Specific Chronic Condition Such as Chronic Obstructive Pulmonary Disease, Heart Failure, Idiopathic Pulmonary Fibrosis, Cystic Fibrosis, Neuromuscular Disease, etc.?

Of the respondents that provided a ‘usual explanation’, fifty-three indicated that they would change an aspect of the explanation (eight indicated they would not). The majority of responses indicated the main change would be specific tailoring of the explanation to the pathophysiology of a disease (35 respondents, 66%, Appendix A). Where the response included differential phrasing for breathlessness associated with specific diseases, common terms/phrases for chronic obstructive pulmonary disease included “difficulty getting air out”, “trapped air”, “hyperinflation”; for interstitial lung disease, “thickened”, “stiff”, “scarred” “can’t get air in”, “restricted”; and heart failure, “heart not pumping well”, “fluid buildup in lungs”, “impaired/reduced oxygen exchange”.

### 3.3. What Do You Think Are the Most Important Points to Convey When Explaining Chronic Breathlessness to a Person with This Symptom?

Fifty-seven respondents provided responses concerning important points to include when explaining chronic breathlessness (Appendix A). Figure 2 presents a summary of the frequency of content domains. Three domains were evident for more than a third of respondents; ‘Management principles’ (63% of respondents), ‘Mechanisms, aggravating and relieving factors’, and ‘Not harmful but persists despite optimal treatment’ (both 37% of respondents). Respectively, the most frequent phrases within these three domains reflected concepts of adapting and self-management despite persistent breathlessness (7 [12%] respondents); the lung pathology and the vicious circles of emotions and behaviors relevant to the individual patient (10 [18%] respondents); and that breathlessness is part of most people’s normal life, is an expected reaction to a situation, but chronic breathlessness is out of proportion to activity, provoked by lesser activities/stress, and feels worse (12 [21%] respondents).

### 3.4. Is There Any Information That You Avoid or Try Not to Say When Explaining Chronic Breathlessness to a Person with This Symptom?

A total of 35 of 59 (59%) respondents providing a response indicated that there was at least one concept they would avoid when explaining chronic breathlessness (Appendix A). The most frequent responses reflected the domain of “Chronic breathlessness is not…” (21 [60%] respondents) with the most frequent concept being that chronic breathlessness is not because oxygen saturation in the blood is definitely low or can be relieved by oxygen as a first-choice treatment option (12 [34%] respondents). The most frequent concepts within the two remaining domains (Inappropriate reassurance/assurance; Blaming and hopelessness, both 8 [23%] respondents) reflected avoiding ‘saying that all breathlessness should be avoided’ (5 [14%] respondents) and ‘saying anything that implies that there is nothing (more) that can be done for it’ (6 [17%] respondents). Specific terms/words to be avoided in explanations offered by individual respondents included “oxygen dependent”, “choking, drowning, suffocating”, “dyspnoea, respiratory failure, respiratory depression”, “intractable”,” untreatable”, “panic”, and “panic disorder”.

### 3.5. When Explaining Chronic Breathlessness to People Living with This Symptom, Are There Particular Ideas or Concepts That You Think People Find Difficult to Understand or Follow?

Of the 59 people providing a response, 49 (83%) indicated that there was at least one concept perceived to be difficult for people living with this symptom to understand (Appendix A). The most frequent was the ‘Inconsistent relationship between oxygenation (SpO_2_, PaO_2_, supplementary oxygen) and breathlessness (19 [39%] respondents). Concepts volunteered by less than 10 respondents concerned relationships between exercise training/inactivity and breathlessness (8 [16%] respondents), pathophysiology of chronic conditions (6 [12%] respondents), and the inability to cure chronic conditions and breathlessness (6 [12%] respondents).

## 4. Discussion

This cross-sectional survey of health professionals provided a snapshot of examples of ‘usual’ explanations, perspectives about important content, and common misunderstandings concerning chronic breathlessness. While the content of explanations varied across respondents, the most frequent generic phrases within a usual explanation described breathlessness due to an individual’s medical condition or impairment of cardiorespiratory function. Where phrases concerning specific physiological mechanisms were included, the most frequent concepts described increased breathing effort to ensure the availability of sufficient oxygen or fuel. Modifications to explanations for specific chronic health conditions reflected the typical pathophysiological features of specific conditions. Within explanations for chronic breathlessness, most respondents indicated that management principles were important to include and phrases encouraging erroneous or maladaptive beliefs were important to avoid. The most frequent concept perceived by respondents to be difficult for people living with chronic breathlessness to understand was the inconsistent relationship between oxygenation and breathlessness.

### 4.1. Why Do Explanations for Chronic Breathlessness Matter?

We could not identify any published empiric studies reporting similarities between maladaptive breathlessness beliefs of health professionals and people living with chronic breathlessness, nor the impact of different chronic breathlessness explanations on individual beliefs, expectations, behaviors, or health outcomes. There are, however, clear conceptual similarities in concurrent maladaptive beliefs of people living with osteoarthritis and health professionals involved in their care. These common shared misconceptions include the nature of relationships between joint damage on images, symptoms, and functional impact, the inevitability of surgical intervention and joint pain as part of normal aging, and concerns about harms associated with regular physical activity [30,31,32]. These misconceptions have the potential to directly inform both treatment advice and engagement with evidence-based recommendations [32].

In theory, explanations about chronic breathlessness provided by credible sources, such as health professionals, have the potential to act as a powerful cognitive input to maintain or modify an individual’s breathlessness beliefs and expectations, and therefore, their overall perception of breathlessness. While there is a growing evidence base for the prevalence, mechanisms, prognostic value of, and interventions for chronic breathlessness [2], in clinical practice, chronic breathlessness remains somewhat ‘invisible’ in terms of priority for specific assessment and management. Serresse et al. (2022) [26] proposed that these ‘invisibilities’ may include an inability to recognize this symptom (patient or health professional), a lack of shared experience or mutual understanding of the experience [empathy gap], and/or an inability to conceive that the breathlessness experience may be detached from objective measurements (maladaptive beliefs).

### 4.2. What Constitutes a ‘Good’ Explanation for Chronic Breathlessness?

For both the person providing the explanation and the recipient of the explanation, a ‘good’ or helpful explanation should align with current breathlessness science, provide a foundation for adaptive beliefs/expectations, address common/individual misconceptions, be coherent, and encourage engagement with evidence-based approaches for breathlessness management [18]. These foundational principles are evident within service models specific to breathlessness intervention services [2,33,34,35]. The data collection process for this study preceded the publication of an international Delphi consensus survey of clinician–researcher experts concerning key concepts to include or avoid when explaining chronic breathlessness to a person living with this symptom [18]. Amongst the concepts reaching consensus as important to include within explanations was the need to emphasize that this symptom is multifactorial and not just due to the underlying medical condition [18].

Within this snapshot of usual explanations, there were clear examples that proposed the multifactorial origins of chronic breathlessness (Table 2 IDs 10, 35, 56, 82) as well as explanations that focused on a singular physiological origin or mechanism (Table 2 IDs 3, 30, 88). Within the current study, the inclusion of causal phrases within breathlessness explanations linking the experience of unpleasant breathlessness with biological impairments of chronic conditions were common. In ‘usual’ explanations for chronic breathlessness, the commonest phrases concerned an illness-based cause (70% of respondents) or physiological mechanisms (44% of respondents) with an explanation logic that appeared to present a “Normal function → impaired function → breathlessness related to impaired normal function” schema. While less than a third of respondents included phrases reflecting factors beyond these within their ‘usual’ breathlessness explanation (Question 1), a greater number of respondents proffered phrases indicating an awareness of the multifactorial nature of the experience of chronic breathlessness when invited to volunteer information that was important to include in explanations (Question 3). For example, 37% (*n* = 21) of respondents provided at least one potential non-disease-specific contributing factor (‘Mechanisms, aggravating and relieving factors’) for information important to include in explanations, compared to 25% (*n* = 14) of respondents that included a potential contributing factor within their usual explanation. Similarly, while very few respondents included phrases reflecting the inconsistent relationships between breathlessness and oxygenation as part of their ‘usual’ explanation (*n* = 3, 5%), when invited to volunteer information that should be avoided in explanations (Question 4), a greater number of respondents specifically commented on the need to avoid stating or inferring that breathlessness means that the oxygen saturation in the blood is definitely low or can be relieved by oxygen (*n* = 12, 34% of respondents).

### 4.3. How Might Explanation Content Encourage Maladaptive Breathlessness Beliefs in People Living with Chronic Breathlessness?

In the context of stable chronic illness states (non-exacerbated or acutely unwell), previously reported maladaptive chronic breathlessness beliefs include the likelihood of suffocation [36], the need to avoid feeling breathless [16,17,18,36], the potential harm of physical exercise [16,17,27,37], that breathlessness indicates potential bodily harm [16,37], and that nothing further can be done [18]. One further key misconception, highlighted by respondents within this study as the most frequent difficult concept to understand (19 (39%) of respondents), concerns the relationship between breathlessness and oxygen/oxygenation/benefits of supplemental oxygen [16,18,36,37].

Within this sample of usual explanations, there were no verbatim examples of phrasing that explicitly, or could be inferred to, convey the need to avoid breathlessness or physical activity, potential bodily harm signaled by breathlessness, or that nothing further could be done to alleviate breathlessness. The one exception to this concerned a small number of usual explanations where phrases as stated or implied that breathlessness is a result of “needing more oxygen”.

Where usual explanations included the term ‘oxygen’ (14 [24%] respondents, Table 2), the context with the explanation varied between a need for or lack of oxygen as the origin of breathlessness, through to oxygen as one of several potential factors leading to breathlessness. It is possible that this reflected the following: (1) the respondents understanding and beliefs; (2) it was appropriate for the responder’s usual clientele (people with confirmed hypoxemia at rest or minimal exertion); or (3) a form of simplified plain language explanation as a cognitive shortcut or heuristic.

The ability to construct an explanation involves a range of cognitive processes (e.g., attention, content retrieval from memory, information processing, metacognition) which are subject to forms of inherence bias [38]. To minimize the cognitive burden of processing time and the volume of information required to decide or create an explanation, inherence bias occurs when individuals unconsciously rely upon cognitive shortcuts (heuristics) to provide an initial satisfactory (i.e., appropriate for the majority of situations/people) rather than optimal answer (i.e., highly specific to the situation/person), often overusing accessible information and overlooking other relevant (but less accessible) information [38,39]. Heuristic shortcuts are a well-recognized form of cognitive efficiency in clinical settings where, for example, patient decision flow charts/algorithms are provided as a starting point for diagnostic or clinical decision making [40,41,42,43]. Fewer studies explore the nature and influence of heuristics on symptom interpretation/misinterpretation [44]. We could not identify any published studies specifically exploring the presence, use, or impact of cognitive shortcuts within the content of explanations for chronic breathlessness. Cognitive shortcuts have the benefit of reduced cognitive processing time but may come at a cost of increased risk of errors of misattribution [45].

Within the usual explanations including the term ‘oxygen’, there were examples that unequivocally linked breathlessness to the body’s need for oxygen (Table 2 ID 3, 37,42, 52, 88). While these explanations were offered by the minority of respondents in this sample, these respondents were frequently engaged in conversations about breathlessness with people under their care (daily—ID 37; weekly—ID 3,42,52; at least once a month—ID 88). Assuming that these specific explanations reflect a simplified cognitive shortcut, in theory, the explanation logic for usual explanations that infer a positive association between chronic breathlessness and oxygen might be mapped as follows:(1)Normal function of lungs and heart is oxygen transportation;(2)Diseases of lungs/heart impair normal function;(3)Chronic breathlessness results from impairment of oxygen transportation.

While the broad logic statements 1 and 2 are consistently true as both standalone and consequent statements, statement 3 is true only for a smaller proportion of people living with chronic breathlessness and would not occur in isolation of other somatic physiological or psychological factors. In theory, this cognitive shortcut leverages broad biologically focused knowledge of normal cardiorespiratory function, with heuristic overgeneralizing (‘misattributing’) the symptom (chronic breathlessness), through prioritizing the oxygen delivery function of the cardiorespiratory system. Hypothetically, a recipient of this heuristic form of breathlessness explanation may form breathlessness beliefs that associate the symptom of breathlessness with the insufficient availability of oxygen, a need for vigilance for monitoring oxygen levels and/or overestimate the potential value of supplemental oxygen to palliate breathlessness.

### 4.4. Strengths and Limitations

Respondents reflected a range of professional disciplines who frequently explained chronic breathlessness; though the sample is small, positive responder bias is likely and it is unknown whether the responses accurately reflect clinical practice. As a result of these limitations, we did not attempt to explore differences in respondents’ characteristics and responses. We did not specifically ask about the respondent’s clinical context. Where respondents volunteered additional information about their professional discipline, the clinical contexts included acute tertiary (hospital) settings (*n* = 10), palliative care (*n* = 7), community/primary settings (*n* = 9), or could not be determined (*n* = 31). It is likely that the professional discipline and the clinical context will influence respondents’ usual explanations and views of the ‘most important’ information to include/avoid in an explanation about breathlessness. We chose not to provide specific case scenarios to limit responses focused upon differential diagnostic reasoning or speculation about current management. The definition of chronic breathlessness used within this survey omitted the term syndrome as originally proposed [29]. The terminology of chronic breathlessness, especially the use of the term syndrome, has been debated [46,47,48,49,50]. While the data provide a foundation for hypothesis generation, direct causation and associations between key descriptive findings between survey questions cannot be confirmed.

## 5. Conclusions

The examples of chronic breathlessness explanations (content and structure) reported in this study may provide a starting point for clinicians to consider their own ‘usual’ explanations. While explanations are low resource, ubiquitous, foundational interventions, there is a paucity of empiric research concerning the specific explanation content or reproducible protocols for assessing understandability and the helpfulness of explanations. The variety of usual explanations for chronic breathlessness reported by the sample of health professionals in this study reflect the challenges of this “routine” communication where the content foci ranged from multifactorial to disease-orientated single-mechanism origins. Most respondents indicated an understanding of current breathlessness science principles with respect to management and the avoidance of phrases encouraging maladaptive beliefs. Whether there is a relationship between the content of and presence of cognitive shortcuts (heuristics) within explanations provided by health professionals for chronic breathlessness remains to be explored.

## Figures and Tables

**Figure 1 healthcare-12-01813-f001:**
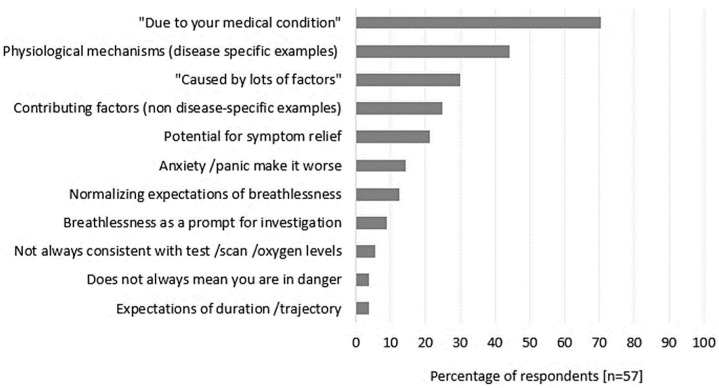
Summary of content domain frequency across respondents for usual explanation for chronic breathlessness. Each respondent’s usual explanation could include more than one domain (mean number of domains per respondent was 2, minimum = 1, maximum = 7).

**Figure 2 healthcare-12-01813-f002:**
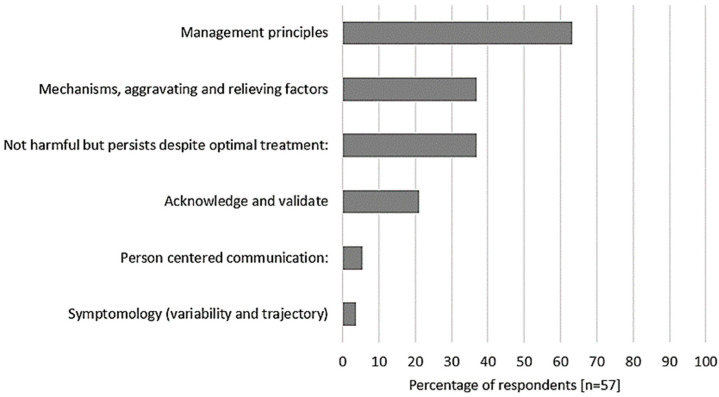
Summary of content domain frequency for important information to include in an explanation for chronic breathlessness.

**Table 1 healthcare-12-01813-t001:** Respondent characteristics.

		N = 61
Professional discipline	Physiotherapy	33
	Nursing	17
	Medicine	10
	Respiratory scientist	1
Role in the past 6 months ^1^	Mainly clinical	47
	Clinical + nonclinical	10
	Nonclinical	4
Frequency of conversations about chronic breathlessness with a patient in the past 6 months	At least once a day	21
	At least once a week	28
	At least once a month	10
	No direct clinical contact	2
“Usual explanation”	Provided text response	61
	Usual explanation provided	57
	Usual explanation not provided ^2^	4
Change usual explanation if person had a specific chronic condition?	Would change	53
	Would not change	8

^1^ Clinical = direct patient contact/management; Nonclinical = academic, research, or administration. ^2^ “Cannot explain until I knew the patient’s condition”.

**Table 2 healthcare-12-01813-t002:** Examples of usual explanations that included the term ‘oxygen’ (unedited complete explanation verbatim). Examples have been arranged according to whether oxygen was the single ‘causative’ factor presented in the explanation (top row), through to oxygen as one of a number of factors causing breathlessness (bottom row).

Responder Identifier	Usual Explanations (Complete Unedited Verbatim Explanation)
ID 3	“You need to work harder with your breathing to make sure your body gets enough oxygen to fuel your muscles”
ID 30	“You have a chronic lung disease which cause air to be trapped. When you feel like you can’t get enough air in, you actually already have too much ‘stale’ air trapped in your lungs, which is why you feel breathless. In order to breath better, you need to focus on lean forward positions to utilise your arm muscles in helping you breathe, pursed lip breath and breath out as much air as possible to get rid of as much of the trapped stale air as possible so that you can get more fresh (oxygenated) air into your lungs.”
ID 37	“Unfortunately, because your lungs are damaged they struggle to produce enough oxygen. Oxygen is like a fuel for the body. Being breathless is your bodies way of saying help, I haven’t got enough fuel for you. Your body thinks that if you breath more you will get more air into your lungs and if not the shortness of breath will slow you down so that the need goes away.
ID 42	“The emphysema has led to a dramatic decrease in the amount of air sacks/alveolus that are able to ‘pick up’ oxygen and get rid of carbon dioxide in the lungs. The air sacks that are remaining are ‘floppy’ and don’t work properly.”
ID 52	“The damage to your airways means that your lungs are not getting air out effectively. This in means that you have less space to get new air in and therefore less oxygenated air getting to your blood. You will therefore be breathing more shallowly and at faster rate.”
ID 54	“Your heart/lungs muscles are a little weaker, so to ensure your body gets the circulation and oxygen it needs, you breathe a little quicker to make sure it gets it.”
ID 88	“Your body is trying to get enough oxygen for all the organs to function as best as possible—by breathing faster this is one ‘shortcut’ to get enough, but it may not the most efficient.”
ID 44	“There are many reasons why people are breathless. Commonly your lungs aren’t able to get in enough oxygen or your muscles are weak, and this makes it harder for your heart and lungs.“
ID 32	“Your lungs aren’t working that well anymore and the tubes that carry the oxygen in your lungs are sometimes tired or not as elastic as they used to be.”
ID 71	“I would explain the need for oxygen when muscles are used to complete daily activities. I would describe changes that have happened to their lungs because of their lung condition (if relevant) and would explain the role of fitness on breathlessness.”
ID 82	“Breathlessness is usually multi-factorial and is most likely related to the way your heart and lungs get oxygen around your body as well as how hard your breathing muscles are working to keep up with the demand for oxygen.”
ID 56	“Breathlessness can be caused from many things and may also be a combination of reasons. It is important to be aware of some of the causes which may be relevant to you. As well, it is helpful to be aware of the things you can do to relieve breathlessness. For example breathing techniques and doing activity in intervals of rest can help manage your breathlessness where possible. Breathlessness is often caused by the inability of the body i.e., the heart and lungs to either use your oxygen properly or pump the blood with oxygen around your body. While breathlessness can be caused from low oxygen levels in the blood this is not necessarily true for some lung diseases. The damage of your lungs or heart may be making it challenging to keep up with the way you breathe during activity. “
ID 10	“Breathlessness is a complex symptom which is the mix of a number of things: your oxygen levels, how good your muscles are at using oxygen, how fit your heart is, and your lung condition.”
ID 35	“There are many reasons you may be breathless; it is not likely to be a single thing that causes it. It may because of the difficulty you have moving air in/out of your lungs, it may be driven be abnormal oxygen or CO2 levels, because of loss of fitness and feelings of worry or panic can make the sensation worse”

## Data Availability

The de-identified raw data supporting the conclusions of this article will be made available by the authors on request.

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
