# Peer review of "Explain Breathlessness: Could ‘Usual’ Explanations Contribute to Maladaptive Beliefs of People Living with Breathlessness?"

_healthcare, 2024, doi:10.3390/healthcare12181813_

Round 1

Reviewer 1 Report

Comments and Suggestions for Authors

This is an interesting snapshot of the explanation provided by health professionals to people living with chronic breathlessness. In methods, the authors could better explain the qualitative analysis used to evaluate the answers of participants. Among the most important points to convey when explaining chronic breathlessness to a person with this symptom are management principles, mechanisms, aggravating and relieving factors, and the notion that it is not harmful but persists despite optimal treatment. Considering the importance of this section, please explain the differences in responses concerning responder characteristics. Presenting the results more clearly would better support conclusions for clinical practice. In the discussion, after explaining "What constitutes a ‘good’ explanation for chronic breathlessness?" and “how the explanation content might encourage maladaptive breathlessness beliefs in people living with chronic breathlessness” please explain the impact. Finally, please explain whether the results of this study are useful in medical practice and in what way.

Author Response

We thank Reviewer 1 for their comments. Our responses to each point are presented in the attached file.

Reviewer 2 Report

Comments and Suggestions for Authors

Greetings Authors
Thank you for your efforts. Although your research sounds clinically significant, I have several inquiries:
1. Your sample size doesn't match the health professional count in Australia and/or New Zealand.
2. Still, I believe physicians should make up the majority of your sample size unless you take into account confounders when presenting your results.
3. I couldn't understand how you included nonpracticing respondents in your sample.

Author Response

We thank Reviewer 2 for their comments. Our responses are presented in the attached file

Reviewer 3 Report

Comments and Suggestions for Authors

General comments:

It is well known that dyspnea is one of the most disabling symptoms for patients and can be a real challenge for clinicians who care these patients. The study's proposal is tremendously interesting as well as brave. On the one hand, there is few literature on the matter and, on the other hand, it is a topic that often has more shadows than lights for teams of health professionals.

This cross-sectional study used an anonymous purpose-designed electronic survey to collect examples of ‘usual’ explanations provided to people living with chronic breath-lessness, and perspectives of explanation content from Australian health professionals.

Weaknesses: Few professionals included in the study (n=61), more than half physiotherapists (n=33) and less than 20% medical doctors (n=10). As described, 57 respondents were finally included.

Perhaps mention should be made that there may be a selection bias. Individuals freely accessed an electronic questionnaire. Perhaps those who dared to answer were "more aware" of the objective theme of this study. More than 90% of respondents to the electronic survey usually give explanations to their patients about this symptom.

Specific comments:

.- Title. “Explain breathlessness”. Maybe it could eliminate this part and leave just the question.

Why do you use the term “breathlessness” and not “dyspnea”?

.- Abstract. Conclusions. As they are written they could be somewhat pretentious. Perhaps you could consider: “This study shows

.- Introduction. Second and third paragraphs start the same way “Explanations are…”. This term is also referred to in the first and fourth paragraph. Please, review and consider the use of any synonym or similar expression.

.- Material and methods. Participants. “Health professionals from a variety of disciplines…”. It could be indicated which disciplines it refers to, e.g., doctors, nurses, physiotherapists, etc.

.- Results. Perhaps the questions could be identified in some way according to the electronic questionnaire number.

.- Line 154. Table 1. “Responder characteristics”. Maybe it would be better “Respondent characteristics”.

.- Table 2. Examples ordered by oxygen levels as the sole or key factor in breathlessness to oxygen? This issue is not well understood. Why not ID identifiers could be placed in numerical order: ID 3, ID 10, ID30, ID 32, etc.

.- Discussion. 

Perhaps the questions posed (in shouts) could be numbered.

Page 9. The first paragraph is a description of results without contrasting them with other studies or justifying these results. Please review.

Page 9. How might explanation content encourage maladaptive breathlessness beliefs in peo- ple living with chronic breathlessness?. This section describes what the maladaptive beliefs are and the results obtained in the study. There is a lack of discussion on this matter. In relation to beliefs, they should be described in the introduction and/or in the material and methods section.

Page 10. Heuristic shortcut. Describe the findings of these shortcuts in the universal literature. There is a lack of comparison of results obtained in the manuscript  and what is described in the scientific literature. Please review.

Page 10. Paragraphs two and three. Of the proposed statements made, reference to any bibliographical citation is missing. Review.

.- Conclusions. The first part of the conclusions is generic and non-specific and does not fit the objectives of this study. Please review.

.- References. 46 quotes are included of which 22 (48%) are recent, that is, five years or less from their publication. Consider including any additional recent references.

Author Response

We thank Reviewer 3 for their considered and thoughtful comments. We have presented our responses in the attached file.

Round 2

Reviewer 2 Report

Comments and Suggestions for Authors

Thank you for your response. It seems scientifically adequate and convincing.